# Cinnamon Oil Inhibits *Penicillium expansum* Growth by Disturbing the Carbohydrate Metabolic Process

**DOI:** 10.3390/jof7020123

**Published:** 2021-02-09

**Authors:** Tongfei Lai, Yangying Sun, Yaoyao Liu, Ran Li, Yuanzhi Chen, Ting Zhou

**Affiliations:** 1Research Centre for Plant RNA Signaling, College of Life and Environmental Science, Hangzhou Normal University, Hangzhou 310036, China; laitongfei@hznu.edu.cn (T.L.); mddran@163.com (R.L.); yzchen530@163.com (Y.C.); 2Hangzhou Key Laboratory for Safety of Agricultural Products, College of Life and Environmental Science, Hangzhou Normal University, Hangzhou 310036, China; sunyangying960824@icloud.com (Y.S.); lyyaiyy1991@163.com (Y.L.)

**Keywords:** *Penicillium expansum*, cinnamon oil, proteome, carbohydrate metabolism

## Abstract

*Penicillium expansum* is a major postharvest pathogen that mainly threatens the global pome fruit industry and causes great economic losses annually. In the present study, the antifungal effects and potential mechanism of cinnamon oil against *P. expansum* were investigated. Results indicated that 0.25 mg L^−1^ cinnamon oil could efficiently inhibit the spore germination, conidial production, mycelial accumulation, and expansion of *P. expansum*. In addition, it could effectively control blue mold rots induced by *P. expansum* in apples. Cinnamon oil could also reduce the expression of genes involved in patulin biosynthesis. Through a proteomic quantitative analysis, a total of 146 differentially expressed proteins (DEPs) involved in the carbohydrate metabolic process, most of which were down-regulated, were noticed for their large number and functional significance. Meanwhile, the expressions of 14 candidate genes corresponding to DEPs and the activities of six key regulatory enzymes (involving in cellulose hydrolyzation, Krebs circle, glycolysis, and pentose phosphate pathway) showed a similar trend in protein levels. In addition, extracellular carbohydrate consumption, intracellular carbohydrate accumulation, and ATP production of *P. expansum* under cinnamon oil stress were significantly decreased. Basing on the correlated and mutually authenticated results, we speculated that disturbing the fungal carbohydrate metabolic process would be partly responsible for the inhibitory effects of cinnamon oil on *P. expansum* growth. The findings would provide new insights into the antimicrobial mode of cinnamon oil.

## 1. Introduction

Food loss and waste by fungal contamination are critical problems in both developing and developed countries. Among the fungi most commonly found, *Penicillium expansum* is the causal agent of blue mold rot disease in a wide range of fresh fruits and vegetables during transport, handling, or postharvest storage. It leads to huge economic losses around the world annually. Additionally, *P. expansum* can produce an array of secondary metabolites such as patulin, citrinin, roquefortine C, chaetoglobosins A and C, and communesins [1,2]. Direct exposure to these mycotoxins will result in serious health problems in humans due to their carcinogenic, immunosuppressive, nephrotoxic, teratogenic, and mutagenic attributes [3,4]. In addition to screening resistant cultivars, blue mold rots can be prevented by careful harvest and handling practices, stringent sanitation, and proper storage conditions. Furthermore, many advances have been made toward chemical and biological control [5,6]. However, so far, the control of *P. expansum* mainly relies on synthetic fungicides containing fludioxonil, pyrimethanil, or difenoconazole as specific active ingredients [7,8]. Nevertheless, the excessive application of fungicides is prone to aggravate chemical residue and develop fungicide-resistant strains. Therefore, with the increased concerns for food safety and environmental contamination, investigations of safe, novel, and efficient antifungal substances or alternative strategies are necessary and urgent.

Essential oils extracted from various aromatic herbs are volatile liquids and generally exhibit potent antioxidative and antimicrobial activities. They are generally regarded as safe substances due to their very low toxicity, which makes them interesting additives in the food industry ([9]. Cinnamon (*Cinnamomum zeylanicum* L.) is a traditional herbal medicine that is grown in almost all tropical regions of the world. Cinnamon oil as an important essential oil contains a large number of aromatic compounds, fatty groups, and terpenoids. A total of 37 components have been detected in cinnamon oil, and the main component is trans-cinnamaldehyde [10]. Currently, cinnamon oil has been applied in many fields due to its multiple functions including anti-inflammatory, anti-cancer, anti-oxidative, insecticidal, and antimicrobial actions [11,12,13,14,15,16]. Cinnamon oil could also maintain or improve the quality of postharvest vegetables and fruits such as mangoes, apples, cherries, jujubes, guavas, and cucumbers [17,18,19,20,21,22]. Cinnamon oil has been used to control postharvest fungi *Penicillium* spp., *Aspergillus* spp., *Rhizopus* spp., *Colletotrichum* spp., *Botrytis cinerea*, *Fusarium verticillioides*, and *Alternaria alternata* in film components, in a direct application or in collaboration with other materials [23,24,25,26,27,28,29]. However, the possible antifungal mechanism of cinnamon oil was not fully understood. Effects of cinnamon oil on the global protein level of *P. expansum* had not been reported. 

Therefore, this study was undertaken to examine the effects of cinnamon oil on *P. expansum*. Proteomic changes in *P. expansum* spores induced by cinnamon oil were investigated. Combining with the biochemical and physiological detections, the potential antifungal mechanisms of cinnamon oil against *P. expansum* were subsequently discussed. The findings will provide a good guiding significance in the postharvest utilization of cinnamon oil.

## 2. Materials and Methods

### 2.1. Antifungal Assays of Cinnamon Oil

*P. expansum* Link (CGMDD3.3703) was cultured on potato dextrose agar (PDA) medium at 25 ^o^C for 10 days. Fresh spores were harvested using sterile water and added to potato dextrose broth (PDB) medium with a final concentration of 1.0 × 10^6^ spores mL^−1^. Cinnamon oil (Product NO. A501966, EINECS NO. 283-479-0, CAS NO. 8015-91-6, EC NO. 616-967-2, Sangon, Shanghai, China) was added to the medium with final concentrations 0, 0.05, 0.15, 0.25, and 0.35 mg L^−1^. Tween-80 with a final concentration of 0.2% (*v*/*v*) was used to promote cinnamon oil dissolving [30]. The spore germination ratio was measured by a Nikon DS-Fi1 microscope after 12 h of culturing at 25 °C under 200 rpm shaking condition. After 24, 48, and 72 h of culturing, the fungal mycelia in a 100 mL system were separated by centrifugation. The mycelia were weighed after drying at 60 °C in an oven (Modell 100–800 m, Memmert, Schwabach, Germany). To assess the effect of cinnamon oil on mycelial expansion, a mycelial agar disk was put on the center of a PDA plate containing 0 or 0.25 mg L^−1^ cinnamon oil. The colony sizes were recorded daily, and the morphologies were photographed by a Nikon Coolpix P7100 digital camera. To evaluate the effect of cinnamon oil on the sporulation of *P. expansum*, 100 μL of spore suspension (1.0 × 10^6^ spores mL^−1^) was evenly spread on a 9 cm Petri dish containing 25 mL of PDA with or without 0.25 mg L^−1^ cinnamon oil. It should be aware that cinnamon oil was added into PDA only when the temperature of PDA was lower than 60 °C. After 3 to 12 days of culturing at 25 °C, the Petri dish was washed by 10 mL of sterile water with 0.2% Tween-20. The spore concentration was determined utilizing a Nikon DS-Fi1 microscope and a hemocytometer.

Apples (*Malus domestica* Borkh cv. Red Fuji) without mechanical injury were bought from the local market of Yuhang district, Hangzhou, China. The fruit were soaked in 2% sodium hypochlorite for 2 min, washed by distilled water twice, and air-dried at room temperature [31]. A round hole (0.3 cm in width and depth) was made at the fruit equator by a sterile nail. Then, 10 μL of the spore suspension at 1.0 × 10^5^ spores mL^−1^ with 0 or 0.25 mg L^−1^ cinnamon oil was inoculated into the wound. Apples were stored at 25 °C and 80% humidity, and disease incidence and lesion diameters were measured daily. Each experiment contained three replicates, and each treatment consisted of 10 apples. The experiment was performed twice.

### 2.2. Fluorescent Staining of Spores

Fresh spores were incubated in PDB medium (1.0 × 10^6^ spores mL^−1^) with or without 0.25 mg L^−1^ cinnamon oil for 6 h at 25 °C under 200 rpm shaking condition. The spores were collected by centrifugation, washed by phosphate buffer solution (PBS, 20 mmoL L^−1^, pH 7.4), and stained by 5μmoL L^−1^ fluorescein diacetate (No. A600202, Sangon, Shanghai, China), 5 μmoL L^−1^ MitoTraker Orange (Invitrogen, Carlsbad, CA, USA), 50 mg L^−1^ 4′, 6′-diamidino-2-phenylindole dihydrochloride (DAPI, No. A606584, Sangon, Shanghai, China), and 20 mg L^−1^ propidium iodide (PI, NO. E607306, Sangon, Shanghai, China). For PI staining, half of the cinnamon oil-treated spores were incubated in boiling water for 10 min, which was as a positive control. The staining was performed following the product instruction. Then, stained spores were observed microscopically and photographed by a Nikon Eclipse Ni-U microscope with individual filter sets.

### 2.3. Quantitative Proteomics Analysis Based on iTRAQ

Fresh spores were incubated in PDB medium (1.0 × 10^7^ spores mL^−1^) containing 0 or 0.25 mg L^−1^ cinnamon oil for 6 h at 25 °C under 200 rpm shaking condition. The spores were collected by centrifugation, washed by PBS twice, and quickly frozen in liquid nitrogen. Each treatment contained three replicates, and the whole experiment was repeated. The equal amounts of spores from three replicas were pooled into one sample. Then, the protein extraction and iTRAQ (Isobaric Tags for Relative and Absolute Quantitation) analysis were performed by LC-Bio Technologies (Hangzhou) CO., LTD following the standard operating procedure of technical service. Raw data files were converted into MGF files by Thermo Proteome Discoverer 1.2 (Thermo, Waltham, MA, USA). Proteins identification was conducted by Mascot search engine (Matrix Science, London, UK) against the reference sequence of *P. expansum* (NCBI: txid27334). Fragment and peptide mass tolerance were ± 0.1Da and ± 0.05Da respectively. Mass values and max missed cleavages were set as monoisotopic and 1. iTRAQ8plex (Y), Gln->pyro-Glu (N-term Q) and oxidation (M) were set as acceptable variable modifications. Carbamidomethyl (C), iTRAQ8plex (N-term), and iTRAQ8plex (K) were set as acceptable fixed modifications. Each confident protein contained one or more unique peptides. The differentially expressed proteins (DEPs) were confirmed by the standard that the ration with *p*-value < 0.05 and fold changes of >1.2 or <0.83 (control versus cinnamon oil-treated spores) in both two biological repeats. The Blast2GO program was used for protein functional annotation against the non-redundant protein database (https://www.ncbi.nlm.nih.gov/protein/ (accessed on 1 January 2021)). DEPs were annotated, classified, and grouped by Clusters of Orthologous Groups of proteins (COGs), Kyoto Encyclopedia of Genes and Genomes (KEGG), and Gene Ontology Consortium databases. The detailed information of DEPs is shown in Appendix A.

### 2.4. Expression Analysis of Related Genes by qRT-PCR

Fresh spores were cultured in PDB medium (1.0 × 10^7^ spores mL^−1^) containing 0 or 0.25 mg L^−1^ cinnamon oil for 6, 12, and 18 h at 25 °C under 200 rpm shaking condition. The spores and mycelia were collected by centrifugation, and total RNAs extraction of samples was performed using TRIzol Reagent (Invitrogen, Carlsbad, CA, USA). First-strand cDNA synthesis was conducted by a FastQuant RT Kit (Tiangen Biotech, Beijing, China) following the product instruction. Relative expression levels of genes were determined by quantitative real-time PCR (qRT-PCR) utilizing a CFX96-real Time System (Bio-Rad, Hercules, CA, USA) and 2 × Ultro SYBr mixture (CW Bio, Beijing, China). Specific primer pairs for genes (*PatA*-*PatO*) involved in the patulin biosynthesis were the same as described by Tannous et al. [32]. Specific primer pairs for genes (*PeCO1-PeCO15*) involved in the carbohydrate metabolic process were designed and shown in Appendix A. The SYBR Green fluorescence intensity change and the threshold cycle (*Ct*) over the background were assessed for each reaction. Expression changes of targeted genes were normalized by reference gene *β-tubulin* and measured using the 2^(−^^△Ct)^ analysis method.

### 2.5. Measurement of Enzymatic Activities

Fresh spores were cultured in PDB medium (1.0 × 10^7^ spores L^−1^) containing 0 or 0.25 mg L^−1^ cinnamon oil for 6, 12, and 18 h at 25 °C under 200 rpm shaking condition. Assay kits from Solarbio Life Science were used for measuring enzymatic activities of collected spores and mycelia. The product numbers are BC2560 for *β*-Glucosidase, BC0710 for *α*-ketoglutarate dehydrogenase, BC0740 for hexokinase, BC0540 for pyruvate kinase, BC0530 for 6-phosphofructokinase, and BC0260 for glucose 6-phosphatedehydrogenase. The operating strictly followed the product instructions. 

### 2.6. Measurement of Carbohydrate, ATP and Patulin Content

Fresh spores were cultured in PDB medium (1.0 × 10^7^ spores mL^−1^) containing 0 or 0.25 mg L^−1^ cinnamon oil for 6, 12, and 18 h at 25 °C under 200 rpm shaking condition. The spores and mycelia were collected by centrifugation, and the carbohydrate content of the supernatant was measured utilizing a portable refractometer (LB20T, Suwei, Taiwan, China). The total carbohydrate content of spores and mycelia was determined by a Total Carbohydrate Content Assay kit (BC2710, Solarbio, Beijing, China). The ATP content was measured by an ATP Content Assay Kit (BC0300, Solarbio, Beijing, China) in strict accordance with the product manual. 

Fresh spores were cultured in PDB medium (1.0 × 10^7^ spores mL^−1^) containing 0 or 0.25 mg L^−1^ cinnamon oil for 2, 4, and 6 days at 25 °C under static condition. After centrifugation, the supernatant was filtered through a 0.22 μm syringe filter (NO. F513152, Sangon, Shanghai, China). Patulin content was evaluated by a reversed-phase HPLC (high-performance liquid chromatography) with UV detection (Waters, Milford, MA, USA). A Waters XTerra RP18 column was used with mobile phases (95% water and 5% acetonitrile). The operational parameters were descripted as Zhou et al. [33].

### 2.7. Statistical Analysis

Unless otherwise specified, data were pooled across three independent repeated experiments. Statistical analysis was conducted by Microsoft Excel software. Analysis of variance was used for comparing more than two means. Mean separations were analyzed by Duncan’s multiple range test. Differences at *p* < 0.05 were considered to be significant. 

## 3. Results

### 3.1. Antifungal Activity of Cinnamon Oil

The antifungal ability of cinnamon oil on *P. expansum* showed a dose-dependent manner. It did not affect the spore germination of *P. expansum* with a concentration of 0.05 mg L^−1^. When the concentration reached 0.25 mg L^−1^, cinnamon oil showed a stable inhibitory effect. After 12 h of treatment, the spore germination rate was below 20%, while it was over 75% in control. With concentration increasing, the inhibitory effect was more positive (Figure 1A). With the extension of culturing, mycelial biomass production of *P. expansum* was significantly lower than that of control in PDB supplemented with 0.25 mg L^−1^ cinnamon oil (Figure 1B), and the mycelial expansion and sporulation were significantly inhibited on PDA under cinnamon oil stress (Figure 1C,D and Appendix A). For inoculation tests in vivo, decay symptoms were found in all inoculated apples, while lesion diameters in treated apples were significantly smaller than those in the control group (Appendix A). Therefore, 0.25 mg L^−1^ was considered as a minimum effective concentration of cinnamon oil on *P. expansum* and used in the subsequent experiments.

To confirm the inhibitory effect of cinnamon oil on *P. expansum*, DAPI staining was used for microscopic observation. After 6 h of culturing, the average volume of control spores was significantly larger than that of cinnamon oil-treated spores. The nuclei in control spores were also bigger, visually clearer, and more regular in shape (Appendix A). Fluorescein diacetate (FDA) is a cell-permeant esterase substrate that can serve as a viability probe. Upon hydrolysis by intracellular esterases, the product yields fluorescein. After FDA staining, compared with the hypofluorescence in treated spores, the control spores emitted a strong and stable green fluorescence (Figure 2A). That indicated that the cellular viability of *P. expansum* was weakened under cinnamon oil stress. The cell-permeant MitoTracker Orange probe containing a mildly thiol-reactive chloromethyl moiety was used for labeling mitochondria. In cinnamon oil-treated spores, the fluorescent intensity was weaker, which indicated that the number of mitochondria decreased or the mitochondria experienced a loss in membrane potential (Figure 2B). These were consistent with the finding that most of the DEPs involved in energy metabolism such as oxidative phosphorylation were remarkably down-regulated under cinnamon oil stress.

The expression changes of 15 genes involved in patulin production in *P. expansum* under cinnamon oil stress were determined. Compared with the control, after 6 h of culturing under cinnamon oil stress, the expression of *PatA*, *PatD*, *PatE,* and *PatL* were up-regulated, and the others were down-regulated or unchanged. With treatment time increasing, all the genes showed a gradually decreasing trend. After 18 h of treatment, the expression levels of all the genes, except for PatD, were significantly lower than those in the control (Appendix A). The effect of cinnamon oil on patulin production was measured by HPLC method. Along with the extension of culturing time, patulin accumulation significantly increased in the control group. After 6 days, the content of patulin reached about 450 μg L^−1^. However, the production of patulin was significantly inhibited by cinnamon oil treatment, and the patulin content was less than one-third of the control (Appendix A). These results indicated that cinnamon oil can weaken the patulin production capability of *P. expansum*, and it was positively correlated with treatment intensity. 

### 3.2. A Global View of iTRAQ Analysis

After 6 h of treatment, the viability of treated spores was different from control spores. However, except for size, the morphologies of control and treated spores were similar. At this time point, the differential expression proteins between control and treatment samples could more exactly reflect the proteomic changes induced by cinnamon oil, which would be helpful for finding cinnamon oil action sites. Therefore, an iTRAQ-based quantitative proteomic analysis was performed focusing on the proteome alteration of samples after 6 h of culturing under cinnamon oil stress. A total of 22,514 unique peptides and 3598 proteins were identified and annotated in all samples. According to the screening threshold, 873 proteins (about 24.3% of the total) were considered as differentially expressed proteins. Among them, 343 DEPs were up-regulated and 530 DEPs were down-regulated. The ratio range of DEPs in expression level was 0.203 to 6.941 (Figure 3). A sum of 159 DEPs that were hypothetical, uncharacterized, or putative proteins were functionally unknown (Appendix A). According to the GO enrichment statistics, most of the functional DEPs with peptidase and organic substance catabolic activities were involved in intracellular components biosynthesis. Through KEGG enrichment analysis, the majority of DEPs participated in butanoate metabolism, fructose and mannose metabolism, ribosome biogenesis, proteasome functioning, linoleic acid metabolism, and glutathione metabolism. The groups involved in the synthesis and degradation of ketone bodies, proteasome, and glutathione metabolism had a higher rich factor value (Figure 4).

It was noteworthy that a total of 146 DEPs were involved in the carbohydrate metabolic process. Specifically, there were 29 for butanoate metabolism, 27 for fructose and mannose metabolism, 12 for propanoate metabolism, 12 for glyoxylate and dicarboxylate metabolism, 26 for pyruvate metabolism, 14 for starch and sucrose metabolism, 9 for pentose and glucuronate interconversions, 17 for glycolysis and gluconeogenesis, 6 for pentose phosphate pathway, and 4 for galactose metabolism. Most of them were down-regulated with the ratio range (control/cinnamon oil) of 0.3 to 6.65 (Figure 5).

### 3.3. Effects of CO on Genes Expression Involved in Carbohydrate Metabolic Process

The expression changes of 14 genes in accordance with DEPs involved in the carbohydrate metabolic process were assessed by qRT-PCR. After 6 h of treatment, compared with a significant decrease of all candidates in the protein level, the expression of four genes (*PeCO3*, *PeCO5*, *PeCO6* and *PeCO7*) were unchanged, three genes (*PeCO1*, *PeCO9* and *PeCO11*) were up-regulated, and another seven genes were down-regulated. After 18 h of treatment, the expression of *PeCO13* changed little, the expressions of *PeCO2* and *PeCO11* were respectively up-regulated more than nine and three times, and all the others were down-regulated remarkably (Table 1). These results indicated that the expression trends of candidates in mRNA levels were similar to those in protein levels with culture time growing. The increase in mRNA levels for a few genes may be due to a cellular stress response under an adverse ambient condition or a compensatory reaction. In addition, CreA is a C2H2 finger domain DNA-binding protein which plays critical roles in carbon catabolite repression and patulin synthesis [34,35]. The expression of *CreA* (*PeCO15*) was first up-regulated after 6 h of treatment. With prolongation of treatment time, the expression was down-regulated to one-third of the control (Table 1).

### 3.4. Effects of Cinnamon Oil on Enzymatic Activities Involving in Carbohydrate Metabolic Process

The effects of cinnamon oil on the activities of six enzymes involves in the carbohydrate metabolic process were evaluated. Among them, the *β-glucosidase* (cellobiase) hydrolyzes maltose and cellobiose, and it acts in the last phase of cellulose degradation process [36]. The α-ketoglutarate dehydrogenase is the fourth enzyme of the Krebs cycle performing the decarboxylation of α-ketoglutarate to succinyl-CoA [37]. The hexokinase catalyzes the first step of glucose metabolism, phosphorylating glucose to glucose 6-phosphate [38]. The 6-phosphofructokinase catalyzes one of the rate-limiting steps of the glycolysis, the phosphorylation of fructose 6-phosphate [39]. Pyruvate kinase catalyzes the irreversible conversion of ADP and phosphoenolpyruvate to ATP and pyruvic acid in the last step of glycolysis [40]. Glucose 6-phosphate dehydrogenase is the rate-limiting enzyme in the first step of the pentose phosphate pathway which oxidizes glucose-6-phosphate to 6-phosphogluconate in the presence of NADP^+^ [41]. In control spores, the enzymatic activities of α-ketoglutarate dehydrogenase, hexokinase, pyruvate kinase, and glucose 6-phosphatedehydrogenase showed a gradually increasing trend with the extension of culturing time, while enzymatic activities of β-glucosidase and 6-phosphofructokinase did not show substantial changes (Figure 6). Under cinnamon oil stress, enzymatic activities of hexokinase, pyruvate kinase, and 6-phosphofructokinase were significantly lower than those in control during 24 h of culture (Figure 6C–E). Compared with the control, enzymatic activities of α-ketoglutarate dehydrogenase and glucose 6-phosphatedehydrogenase did not show obvious changes after 6 h of cinnamon oil treatment, while their activities were significantly lower than those in control along with the increase of treatment time (Figure 6B,F). In addition, enzymatic activities of *β*-Glucosidase increased remarkably at first, and they presented a gradually decreasing trend in cinnamon oil-treated spores (Figure 6A). The results indicated that cinnamon oil showed an inhibitory effect on enzymes involved in the carbohydrate metabolic process.

### 3.5. Effects of Cinnamon Oil on Carbohydrate Consumption and ATP Synthesis

Under cinnamon oil stress, the carbohydrate consumption of *P. expansum* was significantly lower than control (Figure 7A). Meanwhile, the carbohydrate production in spores and mycelia was inhibited by cinnamon oil as well. After 18 h of treatment, the mycelial carbohydrate content in cinnamon oil-treated samples was less half than that in control samples (Figure 7B). The crucial function of carbohydrate metabolism is to produce energy by respiration within the inner membrane of mitochondria. The Mito Tracker Orange staining indicated that the mitochondrial number and membrane potential in cinnamon oil-treated spores decreased remarkably. Showing a decreasing trend with time extension in both control and cinnamon oil-treated groups, the ATP content in cinnamon oil-treated samples was significantly lower than that in control at each time point (Figure 7C). These results indicated that cinnamon oil had a notable inhibitory effect on the carbohydrate metabolic process in *P. expansum*.

## 4. Discussion

*P. expansum* is the most popular and economically significant postharvest pathogen, which mainly threatens the pome fruits and derived products. To control this pathogen, apart from an excessive application of fungicides, great advances have been made toward obtaining alternative antifungal strategies. Among them, the use of essential oils and extracts is one of the optional approaches that meets the requirement of high efficiency, environmental protection, and safety. Due to being less toxic, fat soluble, permeable across living cell membranes, easily undergoing degradation, and its acceptance as biocompatible, cinnamon oil is known as a Generally Recognized as Safe (GRAS) compound and is suggested as a potential candidate for chemoprevention against pathogenic problems. It shows broad-spectrum and potent antifungal, antiviral, and antibacterial activities because of containing many active ingredients such as trans-cinnamaldehyde, eugenol, and linalool [42]. Meanwhile, it has a limited impact on the growth of endogenous strains, which avoids an ecological imbalance that could favor the appearance of new fungal strains or other microorganisms producing other toxic compounds [10,43]. These make cinnamon oil ideal for utilization in various postharvest products.

Particularly for *Penicillium* spp., Ryu and Holt found that cinnamon oil could inhibit *P. expansum* growth and patulin accumulation on apples and in apple juice, and the effectiveness was dependent on the application concentration and the apple cultivar treated [44]. Xing et al. determined the minimum inhibitory concentration of cinnamon oil against *P. expansum* was 0.16% (*v*/*v*). Cinnamon oil with a concentration of 2.0% (*v*/*v*) showed complete control of *P. expansum* in Lingwu Long Jujube and Sand Sugar Orange fruits [45]. Jeong et al. found that cinnamon oil had a wide antifungal spectrum against different *Penicillium* fungal contaminants isolated from cheese. Both essential oils extracted from cinnamon leaf and bark with a concentration of 4000 ppm/mm^2^ exerted powerful antifungal activities against *Penicillium* spp. on Appenzeller cheese [46]. Meanwhile, with the development of active food packaging film, the encapsulation of cinnamon oil in Ag^+^/Zn^2+^-permutite can effectively control decay induced by *Penicillium citrinum* in fresh Chinese bayberry fruit [47]. Moreover, cinnamaldehyde and citral combination led to wrinkles and depressions of spores and mycelia, and lower levels of patulin production in *P. expansum* [48]. In this study, the effects of cinnamon oil on the growth and development of *P. expansum* were firstly evaluated. The results indicated that 0.25 mg L^−1^ cinnamon oil could efficiently inhibit the spore germination, mycelial accumulation and expansion, and conidial production of *P. expansum*. Meanwhile, the blue mold rots in apples induced by *P. expansum* were effectively controlled by cinnamon oil. These results were basically consistent with the previous reports [44,45,46]. The minor differences probably derived from different fungal strains and sources of cinnamon oil. 

Weakened fungal development is not always associated with the decrease of mycotoxin accumulation, since they might have different action modes or a negative feedback relationship. For cinnamon oil, it can play a decisive role in the inhibition of mycotoxin produced by *Fusarium graminearum* in rice culture [49] and the ochratoxin A production of *Aspergillus carbonaria* in apples and pears [50]. Patulin is an important mycotoxin produced by *P. expansum*, and its biosynthesis pathway has been illuminated [32]. The expressions of most genes (except for *PatD* and *PatL*) related to patulin biosynthesis were down-regulated under cinnamon oil stress, which were similar to the previous observation [48]. *PatD* encoded an alcohol dehydrogenase that catalyzed the oxidation and reduction of a wide variety of alcohols and aldehydes. The up-regulation of *PatD* probably played a general detoxifying role under the context of this study. *PatL* encoded C6 transcription factor, which can affect the expression of many genes besides the patulin biosynthetic gene cluster [51]. The previous studies showed that the expression of *PatL* increased under the stress circumstance [52]. Therefore, according to the qRT-PCR and HPLC results, cinnamon oil created a patulin-restrictive condition to *P. expansum*.

Cinnamon oil contains a broad-spectrum antimicrobial effect and has been widely applied in medical and food industries for a long time [10]. Many advances about the antimicrobial mechanism of cinnamon oil have been achieved. Xing et al. found that cinnamon oil could interfere enzymatic reactions of cell wall synthesis and induce irreversible deleterious morphological and ultrastructural alterations of *Fusarium verticillioides* [24]. Zhang et al. elucidated that cinnamon oil led to the leakage of small electrolytes, causing a rapid increase in the electric conductivity of *Escherichia coli* and *Staphylococcus aureus* [12]. Lyu et al. found that cinnamon oil could damage the membrane fatty acids and proteins of *Shewanella putrefaciens* [53]. He et al. suggested that cinnamon oil could destroy the cell membrane integrity and change the structure of cell membrane of *Colletotrichum acutatum* [26]. The transcriptome analysis indicated that cinnamaldehyde and citral combination probably affected the cellular primary structure and limited the nutrition transport, causing energy metabolism disorder and oxidative stress [48]. Chuesiang et al. found that cinnamon oil could disrupt cell wall structures of pathogens and promote the expulsion of internal cellular material [54]. Huang et al. (2019) revealed that cinnamon oil can damage the macromolecules in cell membranes of fish spoilage bacteria according to a Fourier transform infrared (FT-IR) spectroscopy analysis [55]. Yang et al. discovered that *Klebsiella pneumoniae* exposed to cinnamon oil underwent oxidative stress that eventually disrupts the bacterial membrane possibly via interaction with the phospholipid bilayer [56]. Lee et al. emphasized the toxicity of cinnamon oil against *Raffaelea quercus-mongolicae* and *Rhizoctonia solani* due to reactive oxygen species generation and cell membrane disruption [57]. Lee et al. considered that trans-cinnamaldehyde and salicylaldehyde could lead to the down-regulation of an ATP synthesis-related gene cluster, corrupted iron ion homeostasis, and a corrupted ROS defense mechanism in *Agrobacterium tumefaciens* [16]. Among them, one of the mechanisms for the antimicrobial activity of cinnamon oil has been attributed to the disruption of membranes and cell walls for the high lipophilicity. However, in the present study, under 0.25 mg L^−1^ cinnamon oil stress, *P. expansum* spores could keep the membrane integrity after 6 h of culture, and the cell wall did not noticeably change yet (Appendix A). The difference from previous reports might derive from the lower working concentration and the variety of pathogens. Therefore, more explorations were performed to uncover the potential antifungal mechanism of cinnamon oil. 

A global proteomic analysis was performed to illuminate the biological changes of *P. expansum* spores under 6 h of cinnamon oil stress. Through GO and pathway enrichment statistics, DEPs involved in the carbohydrate metabolic process were noticed by the amount and significance. Carbohydrates including monosaccharides, disaccharides, oligosaccharides, and polysaccharides are known as hydrates of carbon or polyhydroxy aldehydes and ketones. They are composed of a six-membered ring (containing carbon and oxygen) either alone or joined together. Carbohydrates and their derivatives have various important biological roles including major energy storage, protein modification and regulation, the backbone of the DNA or RNA, and essential structural components. The carbohydrate metabolism contains various biochemical pathways for biosynthesis, degradation, interconversion, and transport [58]. In the present study, a total of 146 DEPs (about 16.72% of total) related to carbohydrate metabolism were detected in *P. expansum* spores under cinnamon oil treatment. Most of them were down-regulated except for hydroxymethylglutaryl-coenzyme A synthase, glycosyl transferase, malic enzyme, acetolactate synthase, ribose-phosphate diphosphokinase, formyl transferase, thiolase-like and aldolase-type TIM barrel (DEP3, DEP4, DEP5, DEP47, DEP103, DEP256, DEP263, DEP270, DEP300 and DEP327). That indicated that the carbohydrate metabolic capability of *P. expansum* spores was seriously impaired by cinnamon oil. To verify the results at the protein level, the expressions of 14 candidate genes corresponding to DEPs involved in butanoate metabolism, propanoate metabolism, fructose and mannose metabolism, pyruvate metabolism, pentose phosphate pathway, starch and sucrose metabolism, and glycolysis/gluconeogenesis were determined. The expression trends of candidates in mRNA levels were similar to those in protein levels with treatment time growing. In addition, selecting the most energetically favorable carbon source, known as carbon catabolite repression (CCR), is a survival strategy for microorganisms because it supports the rapid growth and development. In *Aspergillus nidulans*, the CCR pathway was mediated by CreA [59]. CreA was essential for growth on various carbon, nitrogen, and lipid sources, and it played a role in amino acid transport and nitrogen assimilation [60]. In *P. expansum*, the deletion of *Cre*A led to a decreased ability to produce patulin and proteolytic enzymes, and to acidify the environment [35]. In the present study, the expression of *CreA* was significantly down-regulated by cinnamon oil with treatment time increasing. The patulin production was inhibited as well. These results indicated that cinnamon oil represented the potential to impair the secondary metabolites biosynthesis and carbon/nitrogen metabolism of *P. expansum*. Furthermore, the activities of six key regulatory enzymes (β-Glucosidase for cellulose hydrolyzation, α-ketoglutarate dehydrogenase for the tricarboxylic acid circle, 6-phosphofructokinase and pyruvate kinase for glycolysis, hexokinase and glucose 6-phosphate dehydrogenase for pentose phosphate pathway) in carbohydrate metabolism decreased significantly. The changes in intracellular macromolecules were ultimately embodied in the decreasing of extracellular carbohydrate consumption, intracellular carbohydrate accumulation, and ATP production. Under cinnamon oil treatment, the declining fluorescent intensity of MitoTracker Orange probe for labeling mitochondria indicated that the number of mitochondria decreased or the mitochondria experienced a loss in membrane potential. That provided more direct evidence for compromised energy metabolism induced by cinnamon oil.

Data acquired from evaluations at the protein, mRNA, biochemical, and physiological levels are correlated and mutually authenticated. Therefore, we speculated that disturbing the fungal carbohydrate metabolic process would partly be responsible for the antifungal action of cinnamon oil. Further exploration will focus on determining the exact components of cinnamon oil and the integrative information for each active component that should be responsible for the inhibitory effect, screening out the detailed action targets and identifying the direct action mechanism of cinnamon oil on the basis of the present available results. In addition, to facilitate the use and improve the efficacy of cinnamon oil in the practical application, optimizing operation parameters and developing novel synergistic strategies with other active substances or carriers are necessary.

In conclusion, cinnamon oil presents an inhibitory effect on the growth and mycotoxin production of postharvest pathogen *P. expansum* with a dose-dependent manner. The antifungal activity is partially attributed to cinnamon oil-induced carbohydrate metabolic disorder. The results would provide a fundamental understanding of antimicrobial action mode of cinnamon oil.

## Figures and Tables

**Figure 1 jof-07-00123-f001:**
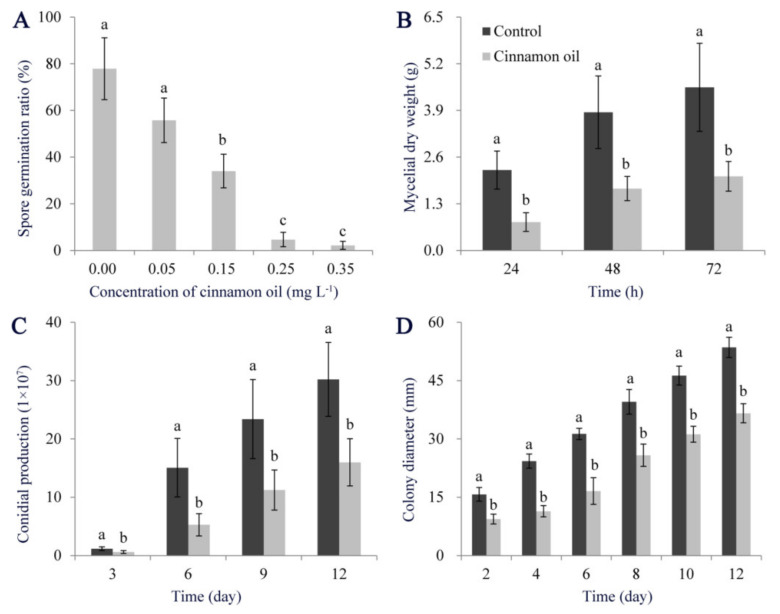
Effects of cinnamon oil on the development of *P. expansum.* Effects of cinnamon oil with different concentrations on spore germination ratio of *P. expansum* after 12 h of culturing (**A**). Effects of cinnamon oil with a concentration of 0.25 mg L^−1^ on hyphae production (**B**), sporulation (**C**), and colony expansion of *P. expansum* (**D**). Bar represents the standard deviation of the means of three independent experiments. Lower-case letters indicate significant differences at *p* < 0.05 at each time point.

**Figure 2 jof-07-00123-f002:**
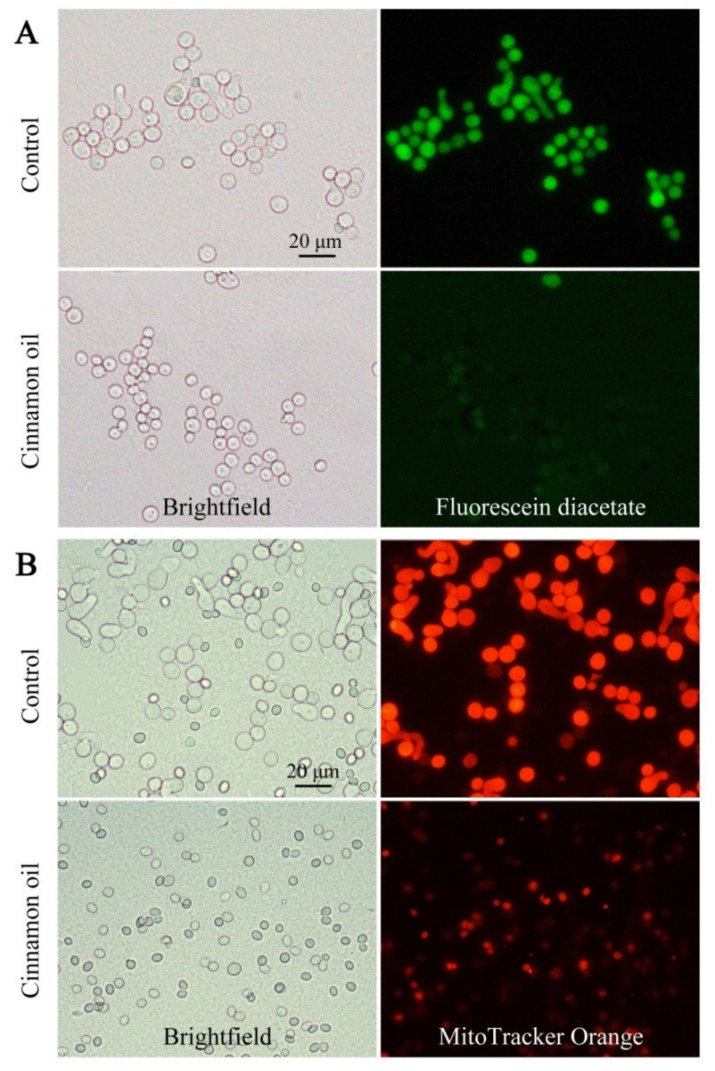
Microscopic observations of spores with fluorescein diacetate (**A**) and MitoTracker Orange staining (**B**) after 6 h of incubation under 0.25 mg L^−1^ cinnamon oil treatment.

**Figure 3 jof-07-00123-f003:**
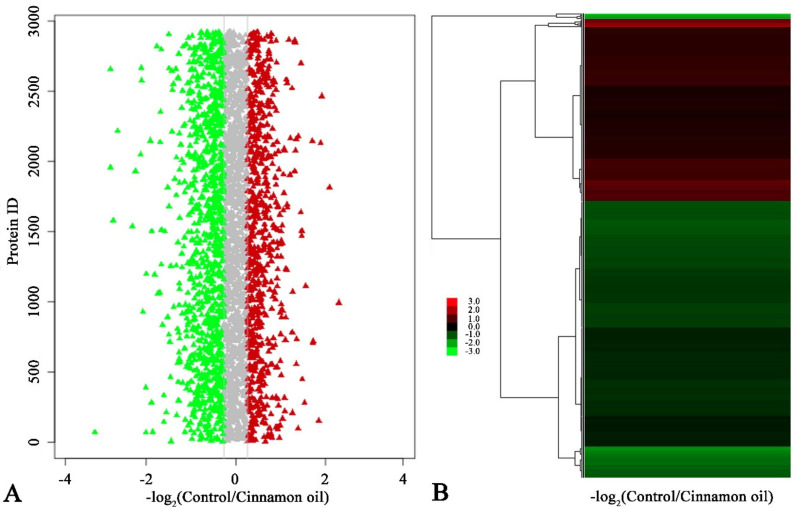
(**A**) Ratio distribution of identified proteins in control and cinnamon oil-treated spores after 6 h of culture. Red, gray, and green triangles represent up-regulated, non-regulated, and down-regulated proteins, respectively. (**B**) Heatmap of the ratio of identified proteins in control and cinnamon oil-treated spores after 6 h of culture. Gradient color barcode (red to green) indicates the up-regulation to down-regulation of protein expression. Each row represents a protein. Proteins with similar fold change values are clustered at the column level. The detailed original data for (**A**,**B**) are listed in Appendix A.

**Figure 4 jof-07-00123-f004:**
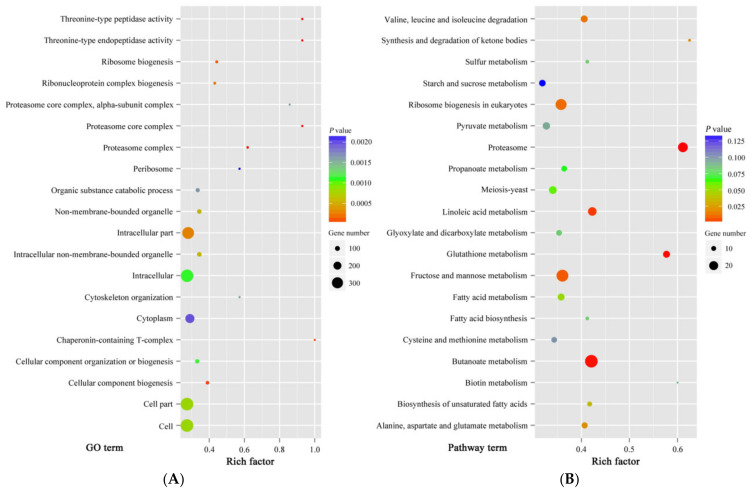
Scatter plots of the top 20 Gene Ontology (GO) (**A**) and Kyoto Encyclopedia of Genes and Genomes (KEGG) (**B**) enrichment of differentially expressed proteins (DEPs) in *P. expansum* spores under 0.25 mg L^−1^ cinnamon oil stress after 6 h of culture. The deeper color in the color code represents the highest confidence of the biological process. Rich factor: the number of DEPs in one GO or KEGG/the number of total identified proteins in the same GO or KEGG. A higher value indicates a higher enrichment level. The size of dot represents the number of DEPs. A larger dot indicates more DEPs.

**Figure 5 jof-07-00123-f005:**
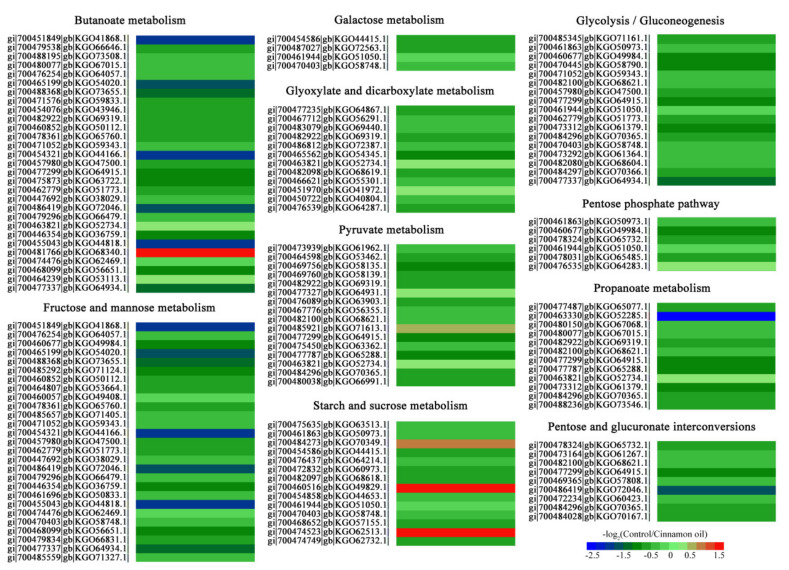
Heatmap of expression changes of DEGs involved in the carbohydrate metabolic process in *P. expansum* spores under 0.25 mg L^−1^ cinnamon oil stress after 6 h of culture. Each row represents a DEP. The annotation and involved metabolic pathway of each DEP are listed. Red to blue in the color code indicates up- to down-regulated expression of the DEPs.

**Figure 6 jof-07-00123-f006:**
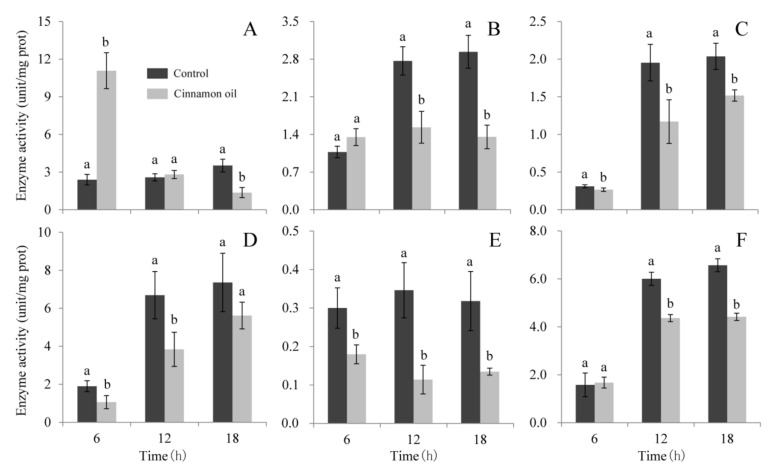
Effects of 0.25 mg L^−1^cinnamon oil on *β*-glucosidase (**A**), *α*-ketoglutarate dehydrogenase (**B**), hexokinase (**C**), pyruvate kinase (**D**), 6-phosphofructokinase (**E**), and glucose 6-phosphate dehydrogenase (**F**) activities in *P. expansum.* One unit of β-glucosidase activity was defined as the amount of enzyme capable of producing 1 nmoL p-nitrophenol per milligram of protein in one hour. One unit of α-ketoglutarate dehydrogenase activity was defined as the amount of enzyme capable of producing 1 nmoL NADH (nicotinamide adenine dinucleotide) per min per milligram of protein. One unit of hexokinase activity was defined as the amount of enzyme capable of producing 1 nmoL NADPH (nicotinamide adenine dinucleotide phosphate) per min per milligram of protein. One unit of pyruvate kinase activity was defined as the amount of enzyme capable of consuming 1 nmoL NADH (nicotinamide adenine dinucleotide) per min per milligram of protein. One unit of 6-phosphofructokinase was defined as the amount of enzyme capable of converting 1 nmoL fructose-6-phosphate and 1 nmoL ATP to 1 nmoL fructose-1, 6-diphosphate and 1 ADP per min per milligram of protein. One unit of 6-phosphatedehydrogenase activity was defined as the amount of enzyme capable of producing 1 nmoL NADPH (nicotinamide adenine dinucleotide phosphate) per min per milligram of protein. Bar represents the standard deviation of the means of three independent experiments. Lower-case letters indicate significant differences at *p* < 0.05 at each time point.

**Figure 7 jof-07-00123-f007:**
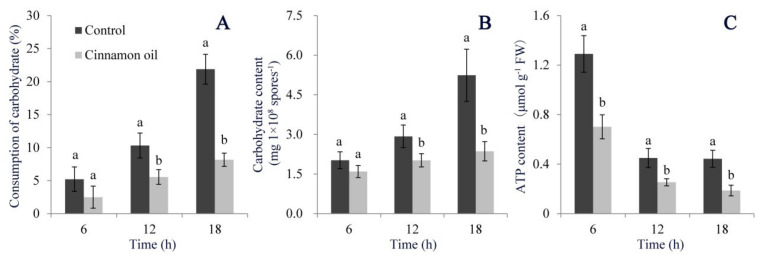
Effects of 0.25 mg L^−1^ cinnamon oil on carbohydrate consumption in potato dextrose broth (PDB) (**A**), intracellular carbohydrate production (**B**), and intracellular ATP content (**C**). Bar represents the standard deviation of the means of three independent experiments. Lower-case letters indicate significant differences at *p* < 0.05 at each time point.

**Table 1 jof-07-00123-t001:** Effects of cinnamon oil on the expression of genes involved in the carbohydrate metabolic process in *P. expansum.*

Genes	6 h		12 h		18 h	
	Control	Cinnamon Oil	Control	Cinnamon Oil	Control	Cinnamon Oil
*PeCO1*	1.01 ± 0.11a	3.56 ± 0.27b	1.01 ± 0.15a	0.95 ± 0.22a	1.01 ± 0.14a	0.84 ± 0.03b
*PeCO2*	1.05 ± 0.35a	0.40 ± 0.14b	1.00 ± 0.17a	3.31 ± 0.32b	1.05 ± 0.36a	9.44 ± 2.62b
*PeCO3*	1.01 ± 0.14a	0.85 ± 0.13a	1.00 ± 0.16a	0.08 ± 0.01b	1.00 ± 0.05a	0.06 ± 0.01b
*PeCO4*	1.01 ± 0.14a	0.67 ± 0.11b	1.01 ± 0.13a	0.05 ± 0.01b	1.02 ± 0.18a	0.04 ± 0.01b
*PeCO5*	1.01 ± 0.16a	1.10 ± 0.36a	1.00 ± 0.10a	0.52 ± 0.10b	1.02 ± 0.22a	0.13 ± 0.02b
*PeCO6*	1.01 ± 0.14a	1.10 ± 0.10a	1.00 ± 0.08a	0.13 ± 0.01b	1.01 ± 0.12a	0.31 ± 0.03b
*PeCO7*	1.02 ± 0.24a	0.95 ± 0.29a	1.00 ± 0.17a	0.62 ± 0.07b	1.01 ± 0.11a	0.07 ± 0.02b
*PeCO8*	1.01 ± 0.14a	0.71 ± 0.10b	1.00 ± 0.09a	0.09 ± 0.01b	1.00 ± 0.06a	0.11 ± 0.03b
*PeCO9*	1.00 ± 0.06a	1.69 ± 0.11b	1.00 ± 0.15a	1.20 ± 0.10a	1.03 ± 0.13a	0.65 ± 0.10b
*PeCO10*	1.00 ± 0.07a	0.67 ± 0.05b	1.00 ± 0.23a	0.63 ± 0.08b	1.01 ± 0.18a	0.33 ± 0.14b
*PeCO11*	1.00 ± 0.09a	3.35 ± 0.60b	1.03 ± 0.25a	4.11 ± 0.74b	1.03 ± 0.24a	3.00 ± 0.49b
*PeCO12*	1.00 ± 0.09a	0.36 ± 0.03b	1.00 ± 0.05a	0.55 ± 0.08b	1.00 ± 0.05a	0.06 ± 0.01b
*PeCO13*	1.00 ± 0.10a	0.71 ± 0.12b	1.00 ± 0.12a	0.99 ± 0.21a	1.01 ± 0.13a	1.09 ± 0.07a
*PeCO14*	1.05 ± 0.27a	0.30 ± 0.08b	1.04 ± 0.31a	0.28 ± 0.17b	1.01 ± 0.18a	0.23 ± 0.05b
*PeCO15*	1.00 ± 0.07a	1.43 ± 0.08b	1.01 ± 0.14a	0.29 ± 0.0.04b	1.00 ± 0.06a	0.37 ± 0.04b

Relative expression levels of *PeCO1* to *PeCO14* in *P. expansum* at different developmental stages are detected by qRT-PCR. The *β-tubulin* is used as the internal control. The detailed information of *PeCO1* to *PeCO14* is shown in Appendix A. Each value is the mean of three independent experiments with the standard deviation. Lower-case letters indicate significant differences at *p* < 0.05 for each gene at the indicated time.

## Data Availability

The data presented in this study are available in article and supplementary material.

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
