# Peer review of "Cinnamon Oil Inhibits Penicillium expansum Growth by Disturbing the Carbohydrate Metabolic Process"

_jof, 2021, doi:10.3390/jof7020123_

Round 1

Reviewer 1 Report

The paper by T. Lai and coworkers is well written, but literature about the antifungal activity of cinnamon oil against P.expansum is not cited nor discussed. The article gives little advancement on the state of the art in the field, a part from the proteomic study performed on cinnamon treated spores, that can be considered as novel. Authors should compare their results with those reported in previous articles and clearly highlight the novelty of their conclusions against the literature before publication be considered.

INTRODUCTION/DISCUSSION: Please, check your results against the literature. Representative articles are listed below:

Yage Xing, Xihong Li, Qinglian Xu, Juan Yun, Yaqing Lu, Antifungal activities of cinnamon oil against Rhizopus nigricans, Aspergillus flavus and Penicillium expansum in vitro and in vivo fruit test, International Journal of Food Science and Technology 2010, 45, 1837–1842. DOI: 10.1111/j.1365-2621.2010.02342.x

Dojin Ryu, Douglas L. Holt, Growth inhibition of Penicillium expansum by several commonly used food ingredients, Journal of Food Protection 1993, 56(10), 862-867. DOI:

Wang Y, Feng K, Yang H, et al Effect of cinnamaldehyde and citral combination on transcriptional profile, growth, oxidative damage and patulin biosynthesis of Penicillium expansum. Front Microbiol 2018, 9, 1–14. DOI: 10.3389/fmicb.2018.00597

Niu B, Yan Z, Shao P, et al, Encapsulation of cinnamon essential oil for active food packaging film with synergistic antimicrobial activity. Nanomaterials 2018, 8, 598. DOI: 10.3390/nano8080598

Eun-Jeong Jeong, Nam Keun Lee, Jisun Oh, Seong Eun Jang, Jai-Sung Lee, In-Hyu Bae, Hyun Hee Oh,Hoo Kil Jung, Yong-Seob Jeong, Inhibitory Effect of Cinnamon Essential Oils on SelectedCheese-contaminating Fungi (Penicillium spp.) during the Cheese-ripening Process, Food Sci. Biotechnol. 2014, 23(4), 1193-1198. DOI 10.1007/s10068-014-0163-8

Discussion (p. 13) should be revised in view of data reported in the abovementioned papers. although previous results obtained against Fusarium graminearum etc. are of interest, authors must primarily compare their results with those reported on the same pathogen (P. expansum).

p.13-14, lines 479-491. Those lines seem to refer to a study conducted by the authors on the production of patulin. If so, they must be moved into the Results section and compared to analogous studies already published in the literature.

p. 14, lines 499-520. Authors refer to the known fungal membrane disruption activity of cinnamon oil. This is somewhat misleading, as it is not at all the only proposed mechanism for the antimicrobial action of cinnamon oil. Papers referring to cinnamon oil ability to affect fungal energy metabolism are of particular relevance to the authors. Please, add a fair list of all the proposed mechanisms and check them against your results.

p. 14, bottom. Please remove lines 530-532.

p. 15, lines 536-541. The text is not sound and unclear. Please, rewrite or delete those lines.

p.15, lines 544-553. Please, delete those lines (it is really puzzling that the authors felt the need to write them in their discussion!)

MINOR POINTS/TYPOS

p.2 line 71: "very low toxicity"

p.9, line 334 (legend to Figure 4): "the highest"

p.10, lines 362-364: "PeCO2 and PeCO11 showed a significantly increasing trend, and all the others were down-regulated remarkably". From Table 1, I cannot see a significant increase in the expression of PeCO11 over time, nor I detect a significant downregulation for all the others. Please correct or explain further.

p.11, table 1: please highlight genes with the highest expression change against control.

p.15, lines 564: "at the protein level"; line 570: "treatment time".; line 576: "changes in"; line 579 "Through assessing in"? Please, rewrite the sentence.

Reviewer 2 Report

Manuscript ID: jof-1092000

Title: Cinnamon oil inhibits Penicillium expansum growth by disturbing the carbohydrate metabolic process

Journal: Journal of Fungi

General comments

This paper deals with the inhibitory effects of cinnamon oil on the growth of P. expansum and the production of its major mycotoxin, patulin. The objective is to find natural compounds to inhibit mycotoxin biosynthesis as an alternative to the use of synthetic fungicides. The authors were interested in the mechanism underlying the inhibitory action of cinnamon oil. They have demonstrated the efficacy on spore germination, conidial production, mycelial growth, reduced expression of genes involved in patulin production, and virulence in vivo. They also found that carbohydrate metabolism could be mainly responsible for the inhibitory effects. The results are clearly presented, illustrations are relevant, and numerous methodologies were used (microscopy, proteomics, measurement of enzymatic activities).

However, some hypotheses would deserve to be better supported by further experimentation and important references are missing in the different sections of the manuscript.

Specific comments

- Please check some form errors, e.g. put species names such as Penicilium expansum in italics.

- Please add a sentence in the introduction or the discussion section:

The authors should keep in mind that cinnamon oil should have a limited impact on the growth of endogenous strains in order to avoid an ecological imbalance that could favor the appearance of new fungal strains or other microorganisms producing other toxic compounds.

In addition, the authors of the study do not however address the antibacterial spectrum of cinnamon oil, although the bacterial ecosystem is known to interact with the fungus development.

- Please add some references:

  • Line 50 (secondary metabolites): Andersen et al 2004 J Agric Food Chem 52: 2421-2428. DOI: 10.1021/jf035406k.
  • Line 53 (toxicity of mycotoxins): Moake et al 2005 Compr Rev Food Sci Food Saf 4: 8-21. DOI: 10.1111/j.1541-4337.2005.tb00068.x
  • Line 137 (apple washing): Sanzani et al 2012 Int J Food Microbiol 153: 323-331. DOI: 10.1016/j.ijfoodmicro.2011.11.021

- Line 129: the addition of cinnamon oil in petri dishes is not described precisely enough. If it is added in a too hot medium, there is a risk of degrading part of the molecules. Please add a sentence to describe the addition of cinnamon oil.

- Line 114: could the authors add the exact composition of cinnamon oil?

- Figure 3 lines 323 and 327: would not there have been an inversion between down- and up-regulation (red=up-regulation and green=down-regulation= usual color code)?

- Lines 288-289: this sentence should be included in the discussion.

-Lines 355-369: a key gene in carbohydrate metabolism is the creA gene. The repression of carbon catabolites (CCR) is a global regulatory mechanism found in a wide range of microorganisms. The CCR pathway is driven by CreA in filamentous fungi and was first described in Aspergillus nidulans. A regulatory effect of CreA on the biosynthesis of secondary metabolites has been observed in some fungal genera. In addition, CreA has been found to affect the pathogenicity of certain plant and animal pathogens.

  • first, the expression change of the creA gene must be tested by qPCR and results included in 3.3 section.
  • results of creA expression should be discussed in addition to the 14 chosen genes with relevant references (e.g. Dowzer and Kelly 1989 Curr Genet. 15, 457-459; Dowzer and Kelly 1991 Mol Cell Biol. 11, 5701-5709; Ries et al 2016) Genetics. 203, 335-352.; Tannous et al 2018, DOI: 10.1111/mpp.12734; and so on).

- Figure 5: the scale is illegible, please enlarge it.

- Lines 490: even if the expression of the genes involved in patulin biosynthesis are down-regulated, leading to a patulin decrease, patulin production should be checked by appropriate assay, e.g. by HPLC, to support this hypothesis.

- Line 590: in perspective, it would be interesting to ask whether one or more particular molecule(s) is/are responsible for the inhibitory effect of cinnamon oil.

- Fig. S2: pictures of apples treated with cinnamon oil versus control are missing.

Reviewer 3 Report

This manuscript shows interesting data about the effect of cinnamon oil on P. expansum from the point of disturbing the carbohydrate metabolism as well as decreasing the patulin production. It is interesting to readers of the journal.

Please check the descriptions carefully again. For example,

L.396, what is "y",

L.505, FI-IR,

L.526, tress.

Author Response

This manuscript shows interesting data about the effect of cinnamon oil on P. expansum from the point of disturbing the carbohydrate metabolism as well as decreasing the patulin production. It is interesting to readers of the journal.

Please check the descriptions carefully again. For example: L.396, what is "y", L.505, FI-IR, L.526, tress.

Response: Thanks for the valuable comments. According to the comments, the revised manuscript has been checked carefully. A colleague who is proficient in scientific English has helped us to check the manuscript in grammar and style.

For example: L.396, what is "y",

The “Y” has been deleted in revised manuscript.

Glucose 6-phosphate dehydrogenase is the rate-limiting enzyme in the first step of the pentose phosphate pathway which oxidizes glucose-6-phosphate to 6-phosphogluconate in the presence of NADP+ (Stanton, 2012). (Lines 435-439 in revised manuscript)

L.505, FI-IR,

The FI-IR has been replaced by FT-IR in revised manuscript.

Huang et al. (2019) revealed that cinnamon oil can damage the macromolecules in cell membranes of fish spoilage bacteria according to FT-IR analysis. (Lines 599-601 in revised manuscript)

L.526, tress.

The “tress” has been replaced by “stress” in revised manuscript.

However, in the present study, under 0.25 mg L-1 cinnamon oil stress, P. expansum spores could keep the membrane integrity after 6 h of culture, and the cell wall did not noticeably change yet (Fig. S5). (Lines 615-618 in revised manuscript)

Finally, we would like to thank you and three reviewers again and we are looking forward to hearing your reply.

Round 2

Reviewer 1 Report

The authors addressed all the issues. I recommend publication of the manuscript.

Reviewer 2 Report

You answered all my remarks and suggestions in a very relevant way. I very much appreciated your cover letter. I have no more remarks to make.